# Clinical Profile and Prognosis of Patients with Left-Sided Infective Endocarditis with Surgical Indication Who Are Not Operated

**DOI:** 10.3390/microorganisms12030607

**Published:** 2024-03-19

**Authors:** María de Miguel, Javier López, Isidre Vilacosta, Carmen Olmos, Carmen Sáez, Gonzalo Cabezón, Pablo Zulet, Adrián Jerónimo, Daniel Gómez, Paloma Pulido, Adrián Lozano, Andrea Oña, Itziar Gómez-Salvador, J. Alberto San Román

**Affiliations:** 1Instituto de Ciencias del Corazón (ICICOR), Hospital Clínico Universitario, CIBER CV, 47003 Valladolid, Spain; javihouston@yahoo.es (J.L.); gonzavilla10@hotmail.com (G.C.); palomapulidogarrido@usal.es (P.P.); lozanoadrian96@gmail.com (A.L.); andreaonaorive@gmail.com (A.O.); itziargs@gmail.com (I.G.-S.); asanroman@secardiologia.es (J.A.S.R.); 2Hospital Clínico San Carlos, CIBER CV, 28040 Madrid, Spain; i.vilacosta@gmail.com (I.V.); pazulet@hotmail.com (P.Z.); adrijeronimo@gmail.com (A.J.); danielgomezramirezmd@gmail.com (D.G.); 3Hospital la Princesa, 28006 Madrid, Spain; csaezbejar@gmail.com

**Keywords:** endocarditis, surgery, mortality

## Abstract

Approximately a quarter of patients with infective endocarditis (IE) who have surgical indication only receive antibiotic treatment. Their short-term prognosis is dismal. We aimed to describe the characteristics of this group of patients to evaluate the mortality according to the cause of rejection and type of surgical indication and to analyze their prognostic factors of mortality. From 2005 to 2022, 1105 patients with definite left-sided IE were consecutively attended in three tertiary hospitals. Of them, 912 (82.5%) had formal surgical indication according to the most recent European Guidelines available in each period of the study and 303 (33%) only received medical treatment. These were older, had more comorbidities and higher in-hospital (46% vs. 24%; *p* < 0.001) and one year mortality (57.1% vs. 27.6%; *p* < 0.001) than operated patients. The main reason for surgical rejection was high surgical risk (57.1%) and the highest mortality when the cause were severe neurological conditions (76%). When the endocarditis team took the decision not to operate (25.5% of the patients), in-hospital (7%) and one-year mortality (17%) were low. In-hospital mortality associated with each surgical indication was 67% in heart failure, 53% in uncontrolled infection and 45% in prevention of embolisms (*p* < 0.001). Heart failure (OR: 2.26 CI95%: 1.29–3.96; *p* = 0.005), *Staphylococcus aureus* (OR: 3.17; CI95%: 1.72–5.86; *p* < 0.001) and persistent infection (OR: 5.07 CI95%: 2.85–9.03) are the independent risk factors of in-hospital mortality. One third of the patients with left-sided IE and formal surgical indication are rejected for surgery. In-hospital mortality is very high, especially when heart failure is the indication for surgery and when severe neurological conditions the reason for rejection. Short term prognosis of patients rejected by a specialized endocarditis team is favorable.

## 1. Introduction

Infective endocarditis (IE) is a severe disease associated with high mortality despite continuous improvements in medical and surgical therapy. This has been attributed to a progressively poorer epidemiological profile during the last decades. Patients are older, have more comorbidities, present more nosocomial episodes, there are more cases of healthcare-related IE, and more prevalence of *Staphylococcus aureus* infection [1]. Antimicrobial treatment is the mainstay of IE therapy, but in most series cardiac surgery is needed in around 50% of patients with left-sided IE (LSIE) [2,3,4,5]. The greatest prognostic benefit of cardiac surgery is achieved in patients at the highest risk.

Practice Guidelines recommend surgery in three main clinical scenarios-heart failure, uncontrolled infection, and prevention of embolisms–generally based on expert consensus and large non-randomized studies [6,7,8,9]. An unneglectable proportion of patients do not undergo surgery despite clear operative indications because the risks outweigh the benefits. Some have very high surgical risk or other conditions that prohibit surgery from being performed. The clinical characteristics, the reasons for refusing surgery, and the predictors of the prognosis of these patients have been barely studied. Besides, previous studies dealing with this subgroup of patients have methodological limitations. Most of them include a few patients [3,5,10,11,12], consider both LSIE and right-sided IE [3], or exclude prosthetic IE [5].

The objectives of our study are: (1) to describe the clinical, microbiological, echocardiographic, and prognostic profile of patients with LSIE rejected for surgery despite having an indication by the European guidelines; (2) to evaluate the prognostic impact according to the cause of rejection and to the type of surgical indication; (3) to determine the prognostic factors of mortality in these patients and (4) to propose possible therapeutic alternatives to surgery.

## 2. Methods

### 2.1. Population

Between 2005 and 2022, every consecutive patient diagnosed of definite LSIE at three tertiary institutions in Spain was included in an ongoing multipurpose database. Of them, we retrospectively selected patients with at least one indication for surgery according to the current European Practice Guidelines in each moment [6,7,8,9]. We excluded those patients who died before surgery or who refused to be operated. Our study is focused on those patients with formal surgical indications by the European Guidelines that did not undergo surgery during the hospitalization in which IE was diagnosed (group A). We have also analyzed patients with IE with surgical indications who were operated (group B) to compare them.

By protocol, patients were interrogated regarding their past and current clinical history and we performed on every patient a physical examination, electrocardiogram, blood and urine analysis, set of 3 blood cultures at admission, 48–72 h later, and at least a transthoracic and transesophageal echocardiography. If blood cultures were negative after 72 h, specific serologic analyses (*Chlamydia*, *Brucella*, *Legionella*, *Mycoplasma*, and *Coxiella*) were obtained. After taking the blood cultures, we started the empirical antibiotic treatment recommended in the guidelines, and afterward, it was modified according to the results of the antibiogram in both groups of patients. The decisions were always shared with an expert in infectious diseases and with the support of the Endocarditis Team (ET).

The protocol conformed to the ethical guidelines of the 1975 Declaration of Helsinki and was approved by the local ethical committees.

### 2.2. The Decision-Making Process

The decision to perform or not to perform surgery was taken by a multidisciplinary IE expert group in each of the participant centers. The ET was composed, at least, by a clinical physician expert in IE, a cardiac surgeon, a cardiologist expert in imaging and a specialist in infectious diseases. Neurologist, geriatricians, nephrologists and other experts were consulted if necessary.

### 2.3. Reasons for Rejection

Patients were rejected for surgery according to three main conditions:Very high surgical risk to the patient and decided by the ET, based on the risk scores available, particularly EUROSCORE II, STS score, RISK-E [13] and the ENDOVAL risk score since 2020. Other conditions which may impact the general prognosis of the patient not considered in the scores (cancer, severe liver disease and/or hemodynamic instability) were also taken into account. Very high surgical risk could also be related to an important structural destruction of the valvular and surrounding tissues not amenable to a surgical repair.Severe neurological conditions, defined as a previous stroke or other neurological diseases with a negative impact on the patient’s daily life, and that shorten his/her life expectancy, or a current hemorrhagic stroke except that which is secondary to a mycotic aneurysm with no clinical sequalae. The rejection for surgery due to this condition was always discussed with a neurologist.Other reasons: Surgical indications accepted in the guidelines have a B or C level of evidence. Thus, some of those indications were not strictly followed. These conditions were not considered an indication for surgery per se: mild heart failure controlled with low doses of diuretics, persistent infection when an alternative antibiotic regimen based on the antibiogram was possible (a specialist in infectious diseases was always consulted in this case) and/or big valvular vegetations without a previous stroke.

### 2.4. Statistics

Continuous variables were reported as mean ± standard deviation or median [interquartile range, IQR] according to variable distribution (normal or not). Student t-test or Mann–Whitney U tests were used for comparing them between groups. The normal distribution of continuous variables was verified with the Kolmogorov–Smirnov test and q-q plot. Categorical variables were reported as absolute values and percentages and compared with a Chi-square test or Fisher’s exact test when expected frequencies were less than 5.

A multivariable analysis by a logistic regression model with the maximum likelihood of using a backward stepwise method was assessed to analyze the prognostic factors of mortality in non-operated patients with surgical indication. All prognostic factors with a *p* value of less than 0.05 in the univariate model were further entered into the multivariate analysis. Age, renal failure, heart failure, septic shock, *Streptococcus viridans*, *Staphylococcus aureus*, persistent infection and periannular complications were included in the model. For all adjusted models, the ratio variable/event was controlled to avoid overfitting. Odds ratios (OR) adjusted for each of the variables included, along with their 95% confidence intervals (95% CI), were calculated. Non-collinearity was checked among the variables. The area under the receiver operating characteristic curve (ROC curve) was used to measure the discriminatory capacity. Calibration was evaluated with the Hosmer–Lemeshow test and with plots. Time to 1-year mortality was analyzed by Kaplan–Meier survival curves, which were compared using the log-rank test.

Statistical analysis was performed using R software, version 3.6.1 (R Project for Statistical Computing). All tests were two-sided. Differences were statistically significant when the *p*-value was <0.05.

## 3. Results

### 3.1. Cohort Identification

After initial inclusion of 1105 definite LSIE patients at the three participant hospitals, 303 met the eligibility criteria for inclusion in the study as having non-operated LSIE, despite formal surgical indication according to the European Practice Guidelines (Figure 1).

### 3.2. Demographic, Clinical, Microbiological, Echocardiographic Characteristics and Outcome of Patients with Surgical Indication Not Operated On

A total of 303 patients did not undergo intervention despite an indication for surgery (group A). They represent 33.2% of the patients with surgical indication of our series. The mean age was 73.2 ± 11.7 years, where 29.0% were older than 70 years, 40.9% were women, 40.0% were referred from other centers and 28.5% had a nosocomial origin. *Staphylococcus aureus* was the most frequent causative microorganism (28.7%), followed by *Enterococci* (19.5%) and coagulase negative *Staphylococci* (17.5%). The majority had echocardiographic vegetations (92.4%) and 27.1% any periannular complications and 59.7% moderate or severe valvular regurgitation. In-hospital and one-year mortality were 46.2% and 57.1%, respectively.

Remarkable differences were found when compared with patients who underwent surgery (group B). The comparison of the baseline main features of both groups is reported in Table 1. Patients from group A were older, more frequently female and had more underlying comorbidities (diabetes mellitus, cancer, chronic renal failure and previous known heart disease). More patients in group A had renal insufficiency or septic shock at admission than in group B. *Staphylococcus aureus* (28.7% vs. 17.0%; *p* < 0.001) and *Enterococci* (19.5% vs. 13.5%; *p* = 0.019) were more frequent in group A, whereas *Streptococcus viridans* was more frequent in group B (10.9% vs. 16.5%; *p* = 0.02). The proportion of methicillin resistant *S. aureus* or coagulase negative *Staphylococci* were similar in both groups. Persistent infection was 37.0% in group A vs. 29.5% in group B (*p* = 0.03). The mean left ventricular eyection fraction were 59.8% in group A and 60.0% en group B (*p* = 0.767). The native mitral valve was affected more often in group A and the aortic in group B. Of note, periannular complications were found in a higher proportion of patients in group B. Differences were found in the median laboratory values of Erythrocyte Sedimentation Rate (ESR) and procalcitonine (PCT). Both logistic Euroscore and Euroscore II were higher in group A (Figure 2). In-hospital mortality was 46% in group A vs. 24% in group B (*p* < 0.001). Figure 3 represents the Kaplan-Meier curve of one year survival in patients operated and rejected for surgery.

### 3.3. Causes of Rejection for Surgery

The reasons for refusing surgery and their respective mortality are shown in Table 2. The main reason for surgical rejection was high surgical risk (57.1%), followed by severe neurological conditions (17.5%). Seventy-seven were not operated on because of the ET decision (25.4%). On the other hand, when severe neurological conditions were the reason for rejection, mortality was the highest. Remarkably, those patients who did not undergo surgery, because the ET had decided otherwise, had a very low mortality (7% during hospitalization and 17% withing the first year of follow-up).

### 3.4. Type of Surgical Indication

The most frequent indication for surgery in group A was uncontrolled infection (176 patients, 58.1%) followed by heart failure (125 patients, 41.3%) and prevention of embolism (142 patients, 46.9%). One hundred and forty patients (46.2%) presented with more than one indication. In-hospital mortality associated with each indication was 67.2% in heart failure, 53.4% in uncontrolled infection and 45.1% in the prevention of embolisms (*p* < 0.001).

### 3.5. Predictors of Mortality in Patients with Surgical Indication Not Operated On

A univariate analysis in the 303 patients with indication for surgery not operated on showed that renal failure, heart failure, septic shock, *Streptococcus viridans*, *Staphylococcus aureus*, persistent infection and periannular complications were statistically associated with mortality (*p* < 0.05). No differences in mortality were found between methicillin re-sistant and non resistant *S. aureus* (66.7% vs. 52.3% respectively; *p* = 0.109) or between methicillin resistant and non resistant coagulase negative *Staphylococci* (34.9% vs. 29.3%; *p* = 0.467).

By multivariable logistic regression analysis, heart failure (OR 2.26; IC95% 1.29–3.96; *p* = 0.005), *S. aureus* (OR 3.17; IC95% 1.72–5.86; *p* < 0.001) and persistent infection (OR 5.07; IC95% 2.85–9.03; *p* < 0.001) remained the only independent predictors of in-hospital mortality. *S. viridans* infection was associated with a lower mortality (OR 0.18; IC95% 0.05–0.66; *p* = 0.009) (Table 3 and Figure 4). The model had an area under ROC curve of 0.792 (CI 95% 0.741–0.844) and a p-valor in Hosmer–Lemershow test of 0.997.

### 3.6. Role of Percutaneous Treatment in Patients Not Operated on Who Have an Indication for Surgery

We have estimated the proportion of patients not operated on which would have been candidates for transcatheter valve replacement (TVR). Patients considered suitable for TVR in the aortic position (TAVR) had to meet all the following criteria: (1) isolated native aortic or biological prosthetic IE; (2) heart failure as the indication for surgery; (3) moderate-to-severe aortic regurgitation or aortic stenosis; (4) absence of periannular complications; and (5) absence of persistent infection.

Similarly, patients considered suitable for TVR in a mitral position had to meet the following: (1) isolated biological prosthetic mitral IE; (2) heart failure as the indication for surgery; (3) moderate-to-severe mitral regurgitation or mitral stenosis; (4) absence of periannular complications; and (5) absence of persistent infection. Out of the 303 patients from group A, 130 had native or bioprosthetic aortic endocarditis, and 13 fulfilled those criteria (10.0%); likewise, 14 patients (4.6%) had a biological mitral prosthesis, and 1 patient fulfilled the criteria.

## 4. Discussion

There are not enough studies in the literature that analyze in detail the characteristics of patients with LSIE who have indication for surgery but do not undergo such procedure for whatever reason, even though they represent a substantial proportion of patients and their prognosis is dismal. The majority of these studies do not take into account relevant aspects like the influence of the surgical indication or the reason for rejection in the outcome and none of them have analyzed the risk factors of mortality of these patients.

There are not enough data regarding patients with LSIE who have indication for surgery but do not undergo such procedure for whatever reason, even though they represent an important proportion of patients.

We have studied one of the largest series to date of this particular group of patients and have obtained several important findings, which may help to better understand this subset of patients to improve their outcome: (1) the clinical, epidemiological, microbiological and echocardiographic profile is different, and the outcome of these patients is worse than those who undergo surgery; (2) mortality is low when the cause of rejection is a shared decision taken by the ET, and lower when the indication is the prevention of embolism; and (3) some patients might benefit from a percutaneous approach. All these findings deserve to be commented on.

In our series, the proportion of patients with a surgical indication (83%) was higher than previously reported [4,5,10,11,12,14], probably reflecting a selection bias due to the characteristics of the participant centers, such as university hospitals that receive a substantial number of patients from other smaller centers without surgical facilities (52% in our series). The proportion of patients rejected for surgery was similar to other reported studies [4,5,10,11,12,14,15], in which it ranges between 24% [4] and 32.5% [15]. The same happens with the comparison of patients operated and rejected: non-operated patients are older, have more underlying comorbidities, worse clinical and microbiological profile and poorer outcomes than those who undergo surgery. Interestingly, female patients are rejected for surgery more often than males, a difference already identified in the series by Chu et al. [4], but not in others [10,11,15]. Bansal A. et al. observed that female sex was associated with a decreased likelihood in undergoing overall cardiac surgery in IE, and that women had higher in-hospital mortality in that context [14]. Our mortality seems to be lower than other studies [3,15,16], but it must be considered that we excluded six patients who died before surgery and eight who refused to be operated on that also died during the same hospitalization.

The American [7] and European Practice Guidelines [17] continue emphasizing the key role of the ET in the diagnosis and management of patients with IE. They reinforce the role of Heart Valve Centers with immediate surgical facilities and an ET to improve the outcome of complicated and uncomplicated IE. Highlighting this, 25% of the patients with surgical indication in our series did not undergo surgery after a careful decision of the multidisciplinary ET decision, and they had favorable outcomes (7% in-hospital mortality, and 17% at 1 year). We do agree with the recommendation that patients with IE have to be treated in centers with extensive experience in the disease, or at least, to establish rapid communication channels between them and the referal centers.

Robust scientific evidence is lacking in the field of IE. In the European Practice Guidelines, 2% of the statements have a level A for recommendation (68 for level C, 51 for level B and 4 for level A). Worldwide experts, some of them co-authors of the guidelines, do not follow the recommendations in many cases [5]. While heart failure and uncontrolled infection are well-established indications of surgery, performing surgery for the prevention of embolisms is more controversial [15]. A high concordance between the ET and the guidelines is found when the indication for surgery is heart failure, but less concordance has been reported in the other two major indications [5]. Our results are in line with the Ramos Martinez study, in which prevention of embolism was less relevant than heart failure and uncontrolled infection for predicting mortality in patients who are refused for surgical treatment [15]. In our opinion, prevention of embolisms is the least robust surgical indication of the guidelines and should be revised. In fact, we have recently seen that patients with large vegetations, but without heart failure or uncontrolled local infection do not benefit from undergoing surgery [18].

Over the years, percutaneous techniques have emerged as a very useful alternative to cardiac surgery in several valvular heart diseases, especially in patients with aortic stenosis and prohibitive [19], or very high, surgical risk [20]. In the context of IE, percutaneous treatment has been traditionally considered as a formal contraindication because of the potential risk of infection of the implanted prosthesis. Our hypothesis is that for inoperable patients with valvular sequelae of the infection, heart failure and controlled infection, TVR could be a good alternative to surgery. With the criteria that we have established, 1 out of 10 of the non-operated patients could benefit from these techniques. In this sense, Santos et al. [21] published a series of 54 patients treated with TAVR after a healed IE. In-hospital and one-year mortality rates were comparable to the control cohort, and there was only one case of IE relapse.

We must acknowledge some limitations of our study. The participant centers are tertiary and approximately half of our patients are referred from other institutions without surgical facilities. This is associated with a doubtless selection bias. The surgical risk evaluation is based on predetermined scales (STS, EUROSCORE, ENDOVAL and RISK-E score) but also on the subjective criteria of the ET [22]. However, the guidelines emphasize the importance of the ET and how it changes the patient prognosis [6,7,8,9].

In summary, one third of the patients with surgical indication in our series do not undergo surgery during the initial admission, mainly because they have a very high surgical risk. A smaller proportion of patients are rejected after a careful ET decision or due to a bad neurological condition. Patients rejected for surgery in our series have almost double the mortality rate than patients operated. However, there are interesting differences depending on the surgical indication and the reason for rejection: when the decision not to operate is taken by an ET and when the indication for surgery is prevention of embolisms, the mortality is much lower. On the opposite, when the neurological condition is the contraindication for surgery, the mortality is the highest. Heart failure, *Staphylococcus aureus* and persistent infection are the independent risk factors of in-hospital mortality in patients rejected for surgery whereas *Streptococcus viridans* infection reduces mortality. Probably in the future, some inoperable selected patients with valvular sequelae of the endocarditis can be treated with per-cutaneous techniques.

## Figures and Tables

**Figure 1 microorganisms-12-00607-f001:**
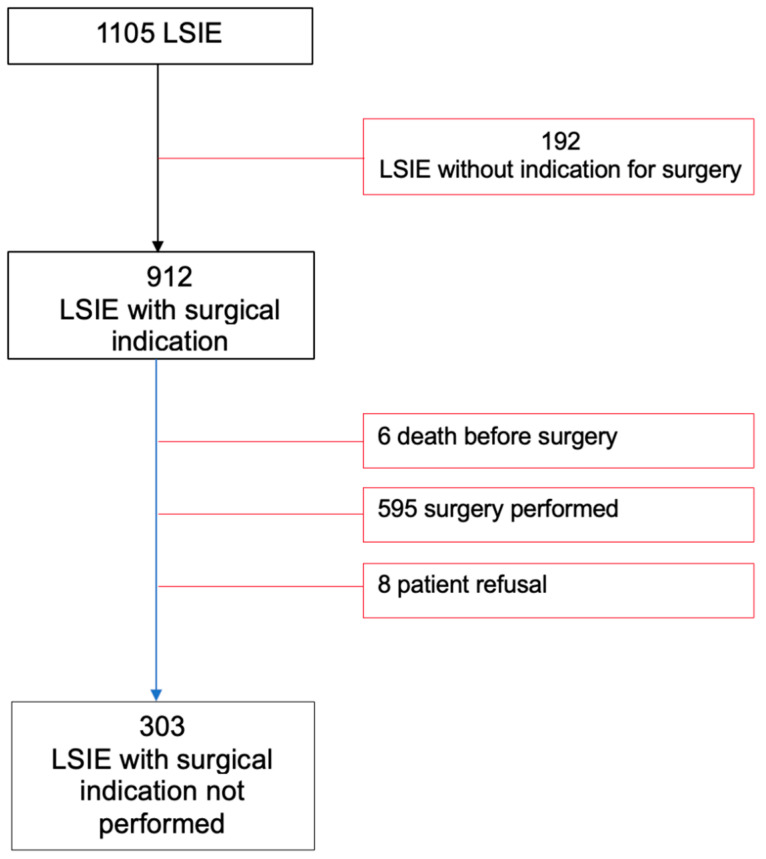
**Eligibility criteria and sample size.**

**Figure 2 microorganisms-12-00607-f002:**
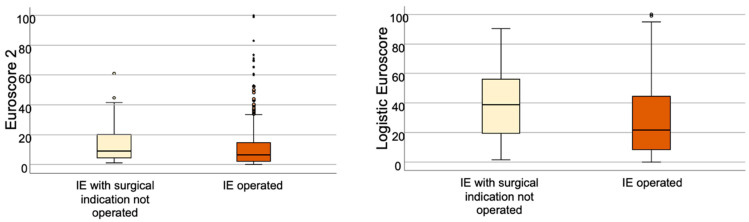
**Euroscore 2 and Logistic Euroscore comparison between patients operated and patients rejected for surgery.**

**Figure 3 microorganisms-12-00607-f003:**
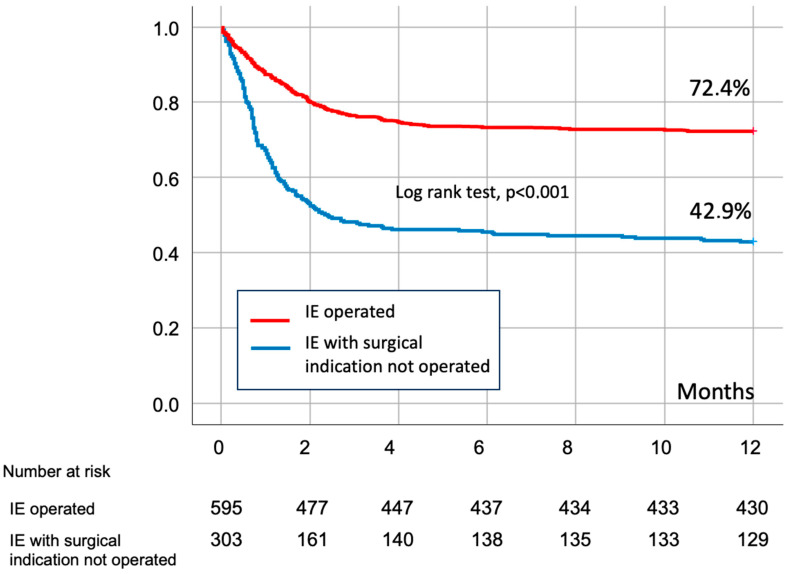
**Kaplan–Meier curve of survival of surgically treated patients and patients refused for surgery.**

**Figure 4 microorganisms-12-00607-f004:**
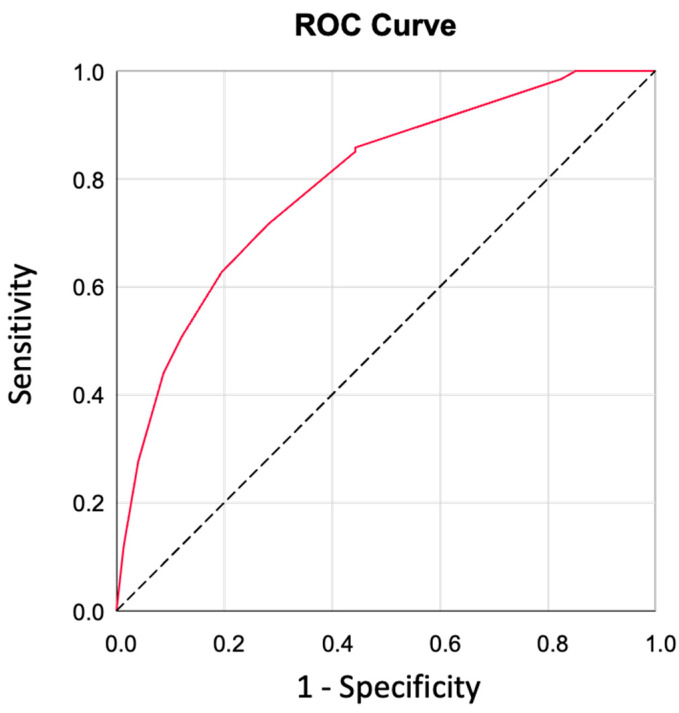
**ROC curve of the multivariate analysis of mortality in patients with surgical indication not operated.**

**Table 1 microorganisms-12-00607-t001:** **Comparison between group A (IE with surgical indication not operated on) and group B (IE operated on).**

VARIABLE	GROUP A IE with Surgical Indication Not Operated (*n* = 303)	GROUP B IE Operated (*n* = 595)	*P*
Epidemiological and clinical characteristics:			
Mean age (years)	73.2 ± 11.7	64.4 ± 13.8	<0.001
<70 years	88 (29.0)	241 (57.3)	<0.001
Female	124 (40.9)	182 (30.6)	0.002
Referred	120 (39.6)	348 (58.5)	<0.001
Nosocomial	86 (28.5)	134 (22.5)	0.050
*Possible port of entry:*			
Dental manipulation	8 (2.6)	43 (7.2)	0.005
Gastrointestinal manipulation	15 (5.0)	28 (4.7)	0.863
Genitourinary manipulation	16 (5.3)	21 (3.5)	0.208
Intravascular catheter	53 (17.5)	68 (11.4)	0.011
**Predisposing conditions** **:**			
Alcoholism	15 (5.0)	50 (8.4)	0.060
Diabetes	99 (32.8)	150 (25.2)	0.017
History of cancer	48 (15.8)	62 (10.4)	0.018
Immunosuppression	28 (9.2)	43 (7.2)	0.284
Cancer	48 (15.9)	62 (10.4)	0.018
Chronic renal failure	73 (24.1)	69 (11.6)	<0.001
Anemia	93 (30.7)	136 (23)	0.010
Previous known heart disease	216 (73.5)	383 (65.9)	0.023
**Mode of presentation:**			
Heart failure at admission	115 (38.9)	268 (45.0)	0.079
Renal insufficiency at admission	109 (36.8)	145 (24.4)	<0.001
Septic shock at admission	40 (13.5)	47 (7.9)	0.008
Positive blood cultures at admission	260 (88.7)	453 (81.0)	0.004
**Microbiological profile:**			
*S. gallolyticus*	11 (3.6)	40 (6.7)	0.058
*Viridans*-group *Streptococci*	33 (10.9)	98 (16.5)	0.025
*Enterococcus spp*	59 (19.5)	80 (13.5)	0.019
Other *Streptococci*	18 (5.9)	39 (6.6)	0.717
*Staphylococcus aureus*	87 (28.7)	101 (17.0)	<0.001
Methicillin resistant	18/87 (20.7)	20/101 (19.8)	0.880
Coagulase negative *Staphylococci*	53 (17.5)	114 (19.2)	0.536
Methicillin resistant	35/53 (66.0)	72/114 (63.2)	0.718
Gram negative bacilli	16 (5.3)	19 (3.2)	0.128
Fungi	3 (1.0)	19 (3.2)	0.043
HACEK	1 (0.3)	3 (0.5)	0.999
Anaerobes	2 (0.7)	16 (2.7)	0.040
Polymicrobial	12 (4.0)	30 (5.0)	0.465
Others	4 (1.3)	20 (3.4)	0.072
Persistent positive blood cultures	74 (38.9)	119 (30.2)	0.035
Negative blood cultures	23 (7.6)	61 (10.3)	0.193
Persistent infection	105 (37.0)	169 (29.5)	0.027
**Echocardiographic characteristics:**			
Vegetation	280 (92.4)	514 (86.5)	0.008
Periannular complication	82 (27.1)	244 (41.0)	<0.001
Valvular stenosis	86 (28.4)	139 (23.4)	0.101
Valvular rupture	17 (5.6)	57 (9.6)	0.041
Moderate-severe regurgitation	181 (59.7)	445 (74.8)	<0.001
Aortic valve	76 (25.1)	226 (38.0)	<0.001
Mitral valve	130 (42.9)	214 (36.0)	0.045
Prosthetic valve EI	134 (44.2)	241 (40.5)	0.285
Native valve IE	188 (62.0)	381 (64.0)	0.559
Moderate to severe pulmonary arterial hypertension (PAH)	195 (64.4)	305 (51.2)	0.094
**Surgical risk and prognosis:**			
Logistic Euroscore	39.1 [19.6–56.2]	21.9 [8.6–44.5]	<0.001
Euroscore II	9.4 [4.6–20.2]	6.6 [2.2–14.8]	0.015
Hospital mortality	140 (46.2)	144 (24.2)	<0.001
One year mortality	173 (57.1)	164 (27.6)	<0.001
**Laboratory *:**			
White blood cell (WBC) count	11,000 [7815–15,655]	10,500 [7695–14,045]	0.314
Erythrocyte sedimentation rate (ESR)	65 [40–94]	55 [36–80.5]	0.018
C-reactive protein (CRP)	16.7 [8.3–56.6]	14.8 [6.2–53.7]	0.206
Procalcitonin (PCT)	1.1 [0.2–27.0]	0.4 [0.1–7.1]	0.016

***** Data are reported as median values.

**Table 2 microorganisms-12-00607-t002:** **Prognostic impact according to the cause of the rejection in 303 non-operated patients.**

	CAUSE OF REJECTION			
	High Risk (*n* = 173)	Neurological Condition (*n* = 53)	ET Decision (*n* = 77)	*p*
In-hospital mortality	95 (55%)	40 (76%)	5 (7%)	<0.001
One-year mortality	116 (67%)	44 (83%)	13 (17%)	<0.001

**Table 3 microorganisms-12-00607-t003:** **Univariate and multivariate analysis of in-hospital mortality in 303 patients rejected for surgery.**

	HOSPITAL MORTALITY			
	UNIVARIATE		MULTIVARIATE	
	OR (95% CI)	*p*-Value	OR (95% CI)	*p*-Value
Age	1.02 (1.01–1.04)	0.023		
Renal failure	1.81 (1.06–3.09)	0.029		
Heart failure	2.67 (1.65–4.32)	<0.001	2.26 (1.29–3.96)	0.005
Septic shock	2.54 (1.19–5.45)	0.017		
*S. viridans*	0.14 (0.05–0.40)	<0.001	0.18 (0.05–0.66)	0.009
*S. aureus*	4.39 (2.55–7.54)	<0.001	3.17 (1.72–5.86)	<0.001
Persistent infection	5.89 (3.45–10.05)	<0.001	5.07 (2.85–9.03)	<0.001
Periannular complications	1.85 (1.11–3.09)	0.019		

## Data Availability

Data are contained within the article.

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
