# Peer review of "Clinical Profile and Prognosis of Patients with Left-Sided Infective Endocarditis with Surgical Indication Who Are Not Operated"

_microorganisms, 2024, doi:10.3390/microorganisms12030607_

Round 1

Reviewer 1 Report

Comments and Suggestions for Authors

I have read with interest the manuscript submitted by De Miguel et al.

I have some comments to be addressed in order to improve the quality of the manuscript:

- if patients were admitted between 2005-2022, how come they had indications for surgery according to the 2015 ESC guidelines?  It should be used, even in a retrospective analysis, the indications that were available at the specific time.

-consider adding more advanced statistical methods (ROC curves, boxplots, etc)

- stating that an article published in 2016 is outdated is a bit harsh, considering that you cited even older articles without considering them outdated.

- in the material and methods section it should be mentioned that the authors divided the patients into 2 groups, not letting the readers figure it out. Moreover, it is stated that only 303 patients met the eligibility criteria for inclusion, and then you proceeded to analyze all cases. 

-table 1 should be organized into categories or divided into smaller tables (with its interpretation); it presents random and disorganized information.

- the conclusions section is missing.

Considering the profile of this journal, consider adding further information about the microorganisms isolated (such as methods of identification - in the material and methods section, antibiotic resistance rates, and antibiotic treatment). 

Moreover,  I suggest the authors give more importance to the aspect of the manuscript, there are multiple fonts and dimensions used, and the reference list is not edited according to the mdpi pattern. Adding some figures would definitely help improve the soundness of the study.

Comments on the Quality of English Language

moderate/extensive editing required.

Reviewer 2 Report

Comments and Suggestions for Authors

The study is potentially interesting, but it lacks detailed statistics. I would suggest including some continuous variables in predicting mortality risk, such as left ventricle ejection fraction, NT-proBNP (and other biomarkers), systolic/diastolic blood pressure at admission, oxygen saturation, and heart rate/rhythm.

Just mentioning the disease (heart failure, sepsis) is not adequate in interpreting the possible confounding factors associated with a poorer outcome.

I also suggest that the manuscript should provide more graphical representations, including concerning the statistics (ROC curves in predicting mortality, and boxplots for certain relevant parameters.

Also, please provide the antibiotic regimens used in deceased patients vs survivors.

Best regards,

The Reviewer

Comments on the Quality of English Language

Review by a native English speaker.

Round 2

Reviewer 1 Report

Comments and Suggestions for Authors

I appreciate the author's efforts in addressing my comments. The quality of the manuscript has improved; still, some issues need to be fixed, such as:

- row 64 - even though the authors mentioned that they used the guidelines available at the time, the reference is only for the 2015 ones. please modify

- there is a mix of presenting percentages either rounded up (especially in the tables) or using one decimal. I highly suggest using a unitary form (all percentages with one additional decimal -> x.y%)

Although the authors provided some information about the microorganism, can they mention if methicillin resistance was associated with higher mortality? How about the antibiotic treatment, in how many cases the empiric treatment was adequate? When a modification was done, the delay had any impact on the outcome of the patients?

- row 194 - please specify that you refer to patients from group A

- please replace "age" from tables with "Mean age (years)"

- rows 255-259 - can the authors comment on this topic? what can be the explanation? 

Conclusions: please check for misspells and expand this section. There are still multiple fonts and text dimensions used 

Comments on the Quality of English Language

several misspells identified.

Reviewer 2 Report

Comments and Suggestions for Authors

The authors only slightly and insufficiently addressed my previous observations:

1)    I do not agree that biomarkers (e.g. NT-proBNP, hs-troponin) are not predictors of a poor outcome in infective endocarditis. Please review the below-mentioned references for evidence:

Concerning NT-proBNP and other biomarkers:

https://pubmed.ncbi.nlm.nih.gov/17493474/

https://pubmed.ncbi.nlm.nih.gov/35430641/

https://www.sciencedirect.com/science/article/pii/S120197121401635X

https://www.sciencedirect.com/science/article/pii/S1201971220301284

Concerning LV ejection fraction and other echo findings:

https://www.ahajournals.org/doi/full/10.1161/CIRCIMAGING.114.003397

https://www.dovepress.com/in-hospital-and-long-term-outcomes-of-infective-endocarditis-in-chroni-peer-reviewed-fulltext-article-IJGM

In order to be suitable for publication, I recommend to include in the statistical analysis at least these two basic parameters, routinely assessed in every patient with cardiovascular pathology (heart failure, infective endocarditis etc).

2)    Confounding factors in patients with heart failure and sepsis don’t refer to the definition of these pathologies, but to certain other conditions that may influence the prognosis such as diabetes mellitus, anemia, cancer, liver cirrhosis, the etiology of sepsis etc. Please add at least 2-3 of these comorbidities (that may also per se enhance the risk for IE).

3-4) The updated figures are fine.

Best regards,

The Reviewer

Comments on the Quality of English Language

Revision by a native English speaker
